# Biologics for Psoriasis during the COVID-19 Pandemic

**DOI:** 10.3390/jcm10071390

**Published:** 2021-03-30

**Authors:** Koji Kamiya, Mayumi Komine, Mamitaro Ohtsuki

**Affiliations:** Department of Dermatology, Jichi Medical University, 3311-1 Yakushiji, Shimotsuke, Tochigi 329-0498, Japan; mkomine12@jichi.ac.jp (M.K.); mamitaro@jichi.ac.jp (M.O.)

**Keywords:** psoriasis, COVID-19, SARS-CoV-2, systemic therapy, biologics

## Abstract

Psoriasis is a chronic, immune-mediated inflammatory disease that predominantly affects the skin and joints. The recent therapeutic development for psoriasis has been remarkable and biologics have dramatically changed the treatment of psoriasis. In moderate-to-severe cases, systemic therapies are required to control their symptoms and biologics can provide greater efficacy when compared with other types of therapies. The coronavirus disease (COVID-19) pandemic has had a great impact on the lives of many people and has worsened substantially worldwide. During the ongoing COVID-19 pandemic, it still remains unclear whether biologics suppress the immune system and increase the risk of COVID-19. In this review, we have summarized the experience with biologics used for treating psoriasis during the COVID-19 pandemic. Biologics seem to be beneficial to COVID-19 infection. Shared decision-making that is based on updated information is highlighted in the time of COVID-19.

## 1. Introduction

Psoriasis is one of the most frequent chronic inflammatory skin diseases [1,2]. In the past decade, various molecular-targeted therapies have been developed, and these therapies have been approved for the treatment of psoriasis [3]. Molecular targeted therapies can be divided into two representative groups: biologics targeting cytokines and receptors involved in psoriasis pathomechanism, and small molecule inhibitors targeting intracellular signaling molecules. Biologics include tumor necrosis factor (TNF), interleukin (IL)-12/23, IL-23, and IL-17 inhibitors; and, small molecule inhibitors include phosphodiesterase-4 (PDE-4) and Janus-activated kinase (JAK) inhibitors. The exacerbation of psoriasis can cause systemic inflammation, leading to cardiovascular comorbidities [4,5]. In psoriatic arthritis (PsA), delays in diagnosis and treatment can cause irreversible joint destruction. Despite early treatment, some patients develop progressive damage and loss of function [6]. Therefore, effective systemic therapies should be considered in moderate-to-severe psoriasis, and biologics can provide significant symptomatic and functional improvement that cannot be achieved with other types of therapies.

Coronavirus disease (COVID-19) is caused by severe acute respiratory syndrome coronavirus 2 (SARS-CoV-2) [7]. The first case was identified in Wuhan, China, in December 2019. This new type of coronavirus has spread uncontrollably to many countries, and a global COVID-19 pandemic is ongoing. Personal protective measures, such as wearing masks, washing hands, alcohol disinfection, social distancing, and staying at home are recommended to prevent infection. In some areas, patients are unable to receive sufficient medical treatment and resources, leading to non-adherence to treatment regimens. Although patients with severe psoriasis require biologics to control disease severity, it is unclear whether biologics suppress the immune system and increase the risk of COVID-19 infection. The current situation may lead to dilemmas regarding the correct choice of treatment. During the COVID-19 pandemic, patients may discontinue treatment, owing to the fear of infection.

Hence, this review describes the effectiveness of biologics for psoriasis during COVID-19 and it discusses the risks and benefits of biologics in the era of the COVID-19 pandemic.

## 2. Risk Factors for the Exacerbation of Psoriasis and COVID-19

The risk factors for the development of psoriasis can be classified into extrinsic and intrinsic factors [8]. Extrinsic risk factors include mechanical stress, drugs, infection, and lifestyle. Vaccination can be also recognized as an extrinsic risk factor and influenza and adenovirus vaccines are often associated with the development of psoriasis [9,10,11]. In contrast, intrinsic risk factors include metabolic syndrome, obesity, diabetes mellitus (DM), dyslipidemia, hypertension, and mental stress [8]. Of these, mechanical stress, certain drugs, infection, and obesity are known to be associated with the exacerbation of psoriasis (Table 1) [12,13,14,15,16,17]. Certain drugs include β-blockers, lithium, anti-malarial drugs, interferons, imiquimod, terbinafine, and anti-programmed cell death-1 (PD-1) monoclonal antibodies [13,17]. Infections, such as streptococcal infection and human immunodeficiency virus (HIV) infection, are well-known risk factors for psoriasis [14,15]. Recently, it has been reported that rhinovirus and coronavirus are the most frequently detected pathogens in acute psoriasis flares after established respiratory virus infection [18]. There have been some case reports regarding the onset and exacerbation of psoriasis after COVID-19 infection [19,20,21,22,23,24].

In contrast, the increased risk of severe course of COVID-19 has not been fully elucidated. In 5700 patients hospitalized with COVID-19 in New York City, hypertension, obesity, and diabetes were the most common comorbidities [25]. As of 23 December 2020, the Centers for Disease Control and Prevention classified the following comorbidities as established risk factors for severe COVID-19: cancer, chronic kidney disease, chronic obstructive pulmonary disease, Down syndrome, heart conditions, immunocompromised state, obesity, pregnancy, sickle cell disease, smoking, and type 2 DM (Table 1) [26,27]. Psoriasis and COVID-19 share obesity as a risk factor for severe illness. However, it is unclear as to whether psoriasis is an important risk factor for severe COVID-19 infections. In a prospective study analyzing the dermatological comorbidity of 93 patients with COVID-19, 17 patients (11 men and six women) were positive for COVID-19 [28]. The most common diseases were superficial fungal infections (five cases, 25%), psoriasis (four cases, 20%), and viral skin diseases (three cases, 15%). In this study, psoriasis was among the most common dermatological diseases. It was speculated that the stress burden caused by the COVID-19 pandemic might have led to an increased number of visits to the outpatient clinic. A recent large epidemiological study showed the association between psoriasis and a higher risk of COVID-19 [29].

## 3. Biologics for Psoriasis

### 3.1. Biologics and COVID-19

Biologics for the treatment of psoriasis inhibit TNF, IL-12/23, IL-23, and IL-17 (Table 2). There are many case reports of psoriasis patients who presented with mild COVID-19 infections during biologic therapy and had favorable outcomes [30,31,32,33,34,35,36,37,38]; however, biologic therapy did not suppress the progression of COVID-19, which resulted in acute respiratory distress syndrome (ARDS) [39]. In COVID-19 patients, higher levels of TNF have been observed [40]. Moreover, the TNF levels were higher in intensive care unit (ICU) patients than in non-ICU patients [40]. SARS-CoV-2 enters host cells via the angiotensin-converting enzyme 2 (ACE2) receptor, which is expressed in various human organs [41]. TNF inhibition might be effective in reducing SARS-CoV-2 infection and the associated organ damage by decreasing TNF-converting enzyme-dependent shedding of the ACE2 ectodomain (Figure 1) [42]. IL-17 appears to be associated with hypercytokinemia in COVID-19 infections. In a patient with severe COVID-19, there was an increased concentration of highly proinflammatory CCR6+ T-helper (Th) 17 cells in the peripheral blood [43]. It is speculated that, in the cytokine storm, the upregulation of IL-17A and possibly IL-17F is mostly responsible for the pathogenesis of COVID-19 and ARDS [44]. IL-17 inhibition might be effective in controlling the cytokine storm due to the correlation between disease severity and the levels of IL-17 and other Th17 cell-related proinflammatory cytokines [45]. In contrast, the role of IL-23 in the pathogenesis of COVID-19 still remains unknown.

### 3.2. At the Beginning of the COVID-19 Pandemic

It has been shown that psoriasis is independently associated with a small, but increased, risk of serious infection, which leads to hospitalization and associated significant morbidity and/or mortality [46]. At the beginning of the COVID-19 pandemic, it was not known whether this was the most appropriate time to commence immunosuppressive therapy in patients with psoriasis [47]. It was suggested that, in areas of high infection rate or outbreaks, treatment with cyclosporine, methotrexate, and TNF inhibitors should be carefully considered, because these drugs have potency to cause immunosuppression [47]. However, immunosuppressive monotherapy, target therapy, and the absence of significant comorbidities could be associated with a lower risk, and a case-by-case assessment seems to be more appropriate than stopping ongoing therapy or reducing therapy in patients with severe psoriasis [48]. Treatment protocols should be prioritized and individualized based on disease severity, other medical conditions, and viral invasiveness [49]. From a rheumatologic point of view, it was suggested to evaluate not only the infectious profile of immunosuppressants, but also the underlying inflammatory nature of psoriatic disease itself, especially if severe and/or associated with articular involvement [50].

At the beginning of the COVID-19 pandemic, there was concern about immunosuppressive or immunomodulating effects that might lead to more susceptible to COVID-19 infections in patients who received biologic therapies [51]. In the pre-coronavirus era, the respiratory infection rates of biologics were similar to those with placebo in phase III trials, and the treatment continuation of biologics was decided based on these data [51]. It has been suggested that patients with psoriasis can continue their treatment during the COVID-19 outbreak, preventing disease flares, because immunosuppressive and immunomodulatory drugs may have the potency to control the cytokine storm that is associated with a poorer outcome in these patients [52]. With due care, biologics for the treatment of psoriasis should not be discontinued during the COVID-19 pandemic [53]. In a retrospective observational study, no hospitalization or death was observed in 980 patients with chronic plaque psoriasis treated with biologics [54]. The limitations of this study include the large difference in sample size between patients and the general population and the very low number of hospitalizations and deaths in the patient group. However, others have suggested that biologic and immunosuppressive therapies in COVID-19 patients should be discontinued and to carefully weigh the risks and benefits of these therapies [55].

Several studies have analyzed the discontinuation of biologics. In a multicenter retrospective study that was conducted during the peak of COVID-19 cases in Canada, 2095 patients received biologic therapy for psoriasis, and the total number of patients who temporarily discontinued their therapy due to COVID-19-related concerns was 23 (1.1%) [56]. In a prospective study that was conducted during the lockdown in Italy, 178 patients were observed, of which 11 (6%) discontinued their therapy due to the lack of safety in continuing [57]. A telephone survey was also conducted during the Italian lockdown period [58]. When 226 patients with negative COVID-19 results were interviewed, 27.9% (63/226) described a worsening of the disease, with 19% (43/226) correlating this to drug withdrawal. The rate of discontinuation varied in different areas [59], possibly due to the regional status of COVID-19 and concerns regarding the increased risk of infection.

### 3.3. During the COVID-19 Pandemic

In a retrospective Italian multicenter observational study, there were no cases of deaths from COVID-related disease in patients with chronic plaque psoriasis treated with biologics [60]. In addition, there was no significant increased risk of hospitalization associated with COVID-related interstitial pneumonia. These observations have been confirmed in other studies [61,62,63,64,65,66,67,68,69]. PsoProtect (Psoriasis Patient Registry for Outcomes, Therapy and Epidemiology of Covid-19 infecTion) is an international web-based registry (www.psoprotect.org) for healthcare providers to report outcomes of COVID-19 in individuals with psoriasis [70]. Based on this registry, the factors that were associated with adverse COVID-19 outcomes were analyzed [71]. A total of 374 patients with confirmed or suspected COVID-19 infection were registered by clinicians from 25 countries. Most of the patients (71%, 267/374) received a biologic therapy, rather than a non-biological systemic agent (18%, 67/374) or no systemic therapy (10%, 36/374). In this registry, 348 patients (93%) fully recovered from COVID-19, 77 (21%) were hospitalized, and nine (2%) died. An increased hospitalization risk was associated with older age, male sex, non-white ethnicity, and comorbid chronic lung disease. Biologics were associated with a lower risk of COVID-19-related hospitalization than non-biologic systemic therapies. Therefore, biologics for the treatment of psoriasis may not be associated with severe COVID-19. Moreover, in a study of a global electronic medical record database, including more than 53 million patient records, the combination of TNF and methotrexate did not increase the risk of hospitalization [72]. In contrast, it was unclear whether biologics are associated with an increased risk of SARS-CoV-2 infection [61,66,69,73]. However, patients with immune-mediated inflammatory diseases, including psoriasis treated with cytokine inhibitors, had reduced susceptibility to SARS-CoV-2 infection when compared with patients not receiving cytokine inhibitors, as well as the general population [74].

PSO-BIO-COVID is an observational, multicentric study, supported by the Italian Society of Dermatology (SIDeMaST), which aimed at evaluating the impact of SARS-CoV-2 infection on the management of patients with psoriasis in Italy during the first year of the pandemic [75]. Patients with moderate-to-severe chronic plaque psoriasis, aged > 18 years, and receiving any biological agent as of 22 February 2020, were enrolled. Of the 12,807 patients with psoriasis, 328 patients (2.6%) stopped treatment during the observation period without consulting their dermatologist, mainly because of the fear of high contagious risk; and, 233 (1.8%) interrupted their therapy after consulting their dermatologist, mainly because of suspected infection or contact with SARS-CoV-2. Therapy continuation during the COVID-19 emergency seems to strictly depend on the quality of information that patients acquire, and only knowledge of epidemiology and preventive measures of COVID-19 prevents biologics discontinuation [76].

### 3.4. Adherence to Treatment

During the COVID-19 pandemic, there has been many restrictions. In some areas, the suspension of all outpatient services was mandated, including clinics for psoriasis patients, and dermatologists had to adapt to provide more counseling to support patients, detect unmet needs, find ways to reassure patients about their disease, and keep them safe at home [77]. The adherence of patients with psoriasis that were treated with systemic therapies was analyzed in an observational single-institution study [78]. A total of 237 patients with psoriasis were interviewed by telephone. In this study, most patients (76.4%) continued to take their medication. However, patients with more than three comorbidities were over six times more likely to not adhere to their treatment. Age, type of treatment, or any particular type of comorbidity did not appear to influence the therapeutic routine. The drug discontinuation seemed to be mainly due to concerns regarding the potential for COVID-19 infection. During the COVID-19 pandemic national lockdown, a multicenter study revealed that treatment safety concerns were significantly more common in patients that were treated with biologics. Of these patients, 40.7% either agreed or strongly agreed to have experienced an increased risk of COVID-19 infection, as compared to 21.3% in the conventional systemic therapy group and 10.9% in the topical therapy group [79]. In a web-based survey in China, 926 questionnaires were collected regarding outdoor activity restriction and income loss [80]. Outdoor activity restriction was positively associated with the exacerbation of psoriasis, stress, and symptoms of anxiety and depression in a dose-response manner, but was not associated with non-adherence. Similarly, income loss was associated with the exacerbation of psoriasis, stress, and symptoms of anxiety and depression. In contrast, income loss was significantly associated with non-adherence to treatment, but it was not associated with healthcare utilization. Non-adherence behavior and perceived stress were independently associated with both income loss and exacerbation of psoriasis. This survey also investigated the association between nonadherence to treatment and patient-reported outcomes of psoriasis [81]. In total, 634 (68.5%) patients reported nonadherence to treatment, and patients that were treated with systemic therapy and topical therapy showed worse adherence than those treated with biologic therapy. Non-adherence to treatment was significantly associated with deterioration of psoriasis, perceived stress, and symptoms of anxiety and depression.

During the COVID-19 pandemic, telemedicine has been one of the most effective strategies for mental health and education of patients, contributing to adherence to treatment. In Taiwan, telemedicine was legally granted by an amendment to Taiwan’s Physician Act in 2018, and its telemedicine service is anticipated to help, not only under-served regions, but also in situations with the COVID-19 pandemic [82]. In Italy, Brunasso et al. started a teledermatology service using telephone and email when lockdown imposed the closure of non-urgent outpatient clinics on 9 March 2020 [83]. Remote consultations included triage for COVID-19 suspected symptoms, an email check of clinical pictures and laboratory examinations, advice for topical and systemic therapy continuation or discontinuation/switch, and rescheduling of the next appointment. This service was effective in preventing an unnecessary worsening of severe chronic skin diseases and poor outcomes due to the withdrawal of current therapy. Furthermore, this service provided an important advantage for female physicians who also took care of their children during lockdown when the schools closed. The limitations of personal dermatological care of patients with skin diseases can be partially compensated by an extension of teledermatology as a convenient and safe method [84].

## 4. Conclusions

In this review, we summarized the risks and benefits of biologics for the treatment of psoriasis during the COVID-19 pandemic. Biologics seem to be beneficial to COVID-19 infection. Vaccines using mRNA technology are expected to prevent the onset and exacerbation of COVID-19 infections. The Psoriasis Group of the Spanish Academy of Dermatology and Venereology and National Psoriasis Foundation developed a series of recommendations and guidance on the management of psoriasis during the COVID-19 pandemic [85,86,87]. Biologics for psoriasis are not a contraindication to COVID-19 mRNA vaccines, and it is recommended that patients with psoriasis should receive COVID-19 mRNA vaccines. There is no evidence that vaccines affect the onset and severity of psoriasis. Registry data should be collected to inform whether COVID-19 vaccines affect the clinical outcomes of psoriasis. During the ongoing pandemic, shared decision-making between clinicians and patients is required based on updated information. This may also change the provision of medical care in the post-COVID-19 era.

## Figures and Tables

**Figure 1 jcm-10-01390-f001:**
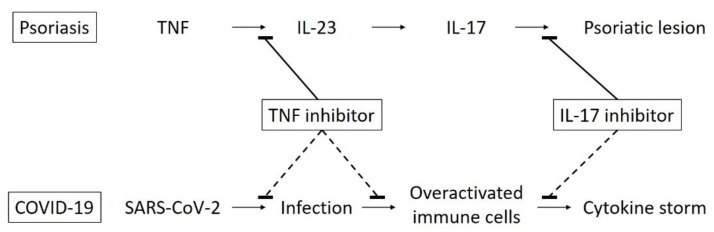
Biologics for psoriasis and COVID-19. Tumor necrosis factor (TNF) inhibitors and interleukin (IL)-17 inhibitors have the potential to prevent the infection of severe acute respiratory syndrome coronavirus 2 (SARS-CoV-2) and the cytokine storm of COVID-19.

**Table 1 jcm-10-01390-t001:** Risk factors for psoriasis and COVID-19.

Psoriasis	COVID-19
Mechanical stress	Cancer
β-blockers	Chronic kidney disease
Lithium	Chronic obstructive pulmonary disease
Anti-malarial drugs	Down syndrome
Interferons	Heart conditions
Imiquimod	Immunocompromised state
Terbinafine	Obesity
Anti-PD1 monoclonal antibodies	Pregnancy
Streptococcal infection	Sickle cell disease
HIV infection	Smoking
Obesity	Type 2 DM

**Table 2 jcm-10-01390-t002:** Biologics for psoriasis.

Classification	Target Molecule	Agent
TNF-inhibitor	TNF	Adalimumab
		Certolizumab pegol
		Etanercept
		Infliximab
IL-12/23 inhibitor	IL-12/23 p40 subunit	Ustekinumab
IL-23 inhibitor	IL-23 p19 subunit	Guselkumab
		Risankizumab
		Tildrakizumab
IL-17 inhibitor	IL-17A	Ixekizumab
		Secukinumab
	IL-17 receptor A	Brodalumab

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
