# Peer review of "Biologics for Psoriasis during the COVID-19 Pandemic"

_jcm, 2021, doi:10.3390/jcm10071390_

Round 1

Reviewer 1 Report

interesting paper.

But the authors should cover the issue on vaccinations regarding to biologics.

For example: Trials to date have not included people taking drugs that affect the immune system and thus the effects of the vaccines in this specific population will need to be established. There is no evidence that vaccines affect psoriasis onset or severity. Registry data should be collected to inform whether SARS-Cov-2 vaccines either positively or negatively affect psoriasis outcomes.

more specific comments:

l52: Extrinsic risk factors include mechanical stress, air pollution, drugs, vaccination, infection, and lifestyle,

Air pollution – reference? Vaccination – reference?

L62: COVID-19 may also be associated with the onset and ex-62 acerbation of psoriasis [16-21].

This has not been established. Maybe for pustular psoriasis, but not for plaque psoriasis.

l78: In this study, psoriasis was among the most common dermatological diseases.

This is just an observation and it could be just that the psoriasis patients were more anxious.

L117-119: It has been suggested that in areas of high infection rate or outbreaks, treatment with cyclosporine, methotrexate, and anti-TNF inhibitors should be carefully considered because these drugs may cause decreased immune response and greater susceptibility to life-threatening infections.

But if the dermatologist stop the treatment of psoriasis patients and their psoriasis worsens, this might increase the risk of COVID-19 due to worsening of the skin, increased stress etc.

L252: There is still uncertainty about the risks and benefits of biologics owing to the quality of the existing data and lack of detailed patient analysis.

I dont agree with this conclusion. Today it can be stated that psoriasis per se, is not a risk factor for SARS-CoV-2 infection or for a more severe COVID-19 disease course.

the authors should include the following article:

Patients with immune-mediated inflammatory diseases receiving cytokine inhibitors have a low prevalence of SARS-CoV-2 seroconversion

Simon D, Tascilar K, Krönke G, Kleyer A, Zaiss MM, Heppt F, Meder C, Atreya R, Klenske E, Dietrich P, Abdullah A, Kliem T, Corte G, Morf H, Leppkes M, Kremer AE, Ramming A, Pachowsky M, Schuch F, Ronneberger M, Kleinert S, Maier C, Hueber AJ, Manger K, Manger B, Berking C, Tenbusch M, Überla K, Sticherling M, Neurath MF, Schett G. Nat Commun. 2020 Jul 24;11(1):3774. doi: 10.1038/s41467-020-17703-6. PMID: 32709909; PMCID: PMC7382482.

These findings led the authors to conclude that patients with IMIDs on cytokine inhibitor therapies have reduced susceptibility to SARS-CoV2 infection compared with patients not receiving cytokine-inhibitors, as well as the general population.

Author Response

The list of answers for reviewer’s comments

  We are grateful for your critical evaluation of our manuscript and for the detailed evaluation. We are also grateful for the contributions obtained from the reviewers and the opportunity to improve the quality of our manuscript. We have made all the suggestions you requested and have revised our manuscript. Additionally, we used colored text in all areas that we changed in the new manuscript to facilitate the revision process.

Comments and Suggestions for Authors

interesting paper.

But the authors should cover the issue on vaccinations regarding to biologics.

For example: Trials to date have not included people taking drugs that affect the immune system and thus the effects of the vaccines in this specific population will need to be established. There is no evidence that vaccines affect psoriasis onset or severity. Registry data should be collected to inform whether SARS-Cov-2 vaccines either positively or negatively affect psoriasis outcomes.

Response: We thank the reviewer for pointing this issue. In the new manuscript, we have incorporated the following sentences as the reviewer kindly suggests; Biologics for psoriasis are not a contraindication to COVID-19 mRNA vaccines and it is recommended that patients with psoriasis should receive COVID-19 mRNA vaccines. There is no evidence that vaccines affect the onset and severity of psoriasis. Registry data should be collected to inform whether COVID-19 vaccines affect the clinical outcomes of psoriasis (line 242).

more specific comments:

l52: Extrinsic risk factors include mechanical stress, air pollution, drugs, vaccination, infection, and lifestyle,

Air pollution – reference? Vaccination – reference?

Response: We thank the reviewer for pointing this issue. Various air pollutants damage the skin by inducing oxidative stress. However, the association between air pollution and psoriasis has not been well established. In the new manuscript, we have excluded air pollution as an extrinsic risk factor. In contrast, vaccination such as influenza and adenovirus is often associated with the development of psoriasis. In the new manuscript, we have incorporated the new references and revised our manuscript as follows; Extrinsic risk factors include mechanical stress, drugs, infection, and lifestyle. Vaccination can be also recognized as an extrinsic risk factor and influenza and adenovirus vaccines are often associated with the development of psoriasis (line 52).

L62: COVID-19 may also be associated with the onset and ex-62 acerbation of psoriasis [16-21].

This has not been established. Maybe for pustular psoriasis, but not for plaque psoriasis.

Response: We thank the reviewer for pointing this issue. As the reviewer pointed out, the association between COVID-19 and the development of psoriasis has not been established. However, there are some case reports about the development of psoriasis after the COVID-19 infection. The type of psoriasis included guttate psoriasis, plaque psoriasis, psoriatic psoriasis, and pustular psoriasis. In the new manuscript, we have revised our manuscript as follows; There have been some case reports about the onset and exacerbation of psoriasis after COVID-19 infection (line 63).

l78: In this study, psoriasis was among the most common dermatological diseases.

This is just an observation and it could be just that the psoriasis patients were more anxious.

Response: We thank the reviewer for pointing this issue. In the new manuscript, we have deleted the following sentences; It was also speculated that immunosuppressive drugs might have been related to the greater number of patients with psoriasis. However, the sample size was too small to confirm the association between psoriasis and COVID-19 (line 81).

L117-119: It has been suggested that in areas of high infection rate or outbreaks, treatment with cyclosporine, methotrexate, and anti-TNF inhibitors should be carefully considered because these drugs may cause decreased immune response and greater susceptibility to life-threatening infections.

But if the dermatologist stop the treatment of psoriasis patients and their psoriasis worsens, this might increase the risk of COVID-19 due to worsening of the skin, increased stress etc.

Response: We thank the reviewer for pointing this issue. We agree with your comments. In the new manuscript, we have revised our manuscript as follows; It was suggested that in areas of high infection rate or outbreaks, treatment with cyclosporine, methotrexate, and TNF inhibitors should be carefully considered because these drugs have potency to cause immunosuppression. However, immunosuppressive monotherapy, target therapy, and absence of significant comorbidities could be associated with a lower risk, and a case-by-case assessment seems more appropriate than stopping ongoing therapy or reducing therapy in patients with severe psoriasis (line 113).

L252: There is still uncertainty about the risks and benefits of biologics owing to the quality of the existing data and lack of detailed patient analysis.

I dont agree with this conclusion. Today it can be stated that psoriasis per se, is not a risk factor for SARS-CoV-2 infection or for a more severe COVID-19 disease course.

the authors should include the following article:

Patients with immune-mediated inflammatory diseases receiving cytokine inhibitors have a low prevalence of SARS-CoV-2 seroconversion

Simon D, Tascilar K, Krönke G, Kleyer A, Zaiss MM, Heppt F, Meder C, Atreya R, Klenske E, Dietrich P, Abdullah A, Kliem T, Corte G, Morf H, Leppkes M, Kremer AE, Ramming A, Pachowsky M, Schuch F, Ronneberger M, Kleinert S, Maier C, Hueber AJ, Manger K, Manger B, Berking C, Tenbusch M, Überla K, Sticherling M, Neurath MF, Schett G. Nat Commun. 2020 Jul 24;11(1):3774. doi: 10.1038/s41467-020-17703-6. PMID: 32709909; PMCID: PMC7382482.

These findings led the authors to conclude that patients with IMIDs on cytokine inhibitor therapies have reduced susceptibility to SARS-CoV2 infection compared with patients not receiving cytokine-inhibitors, as well as the general population.

Response: We thank the reviewer for pointing this issue. As the reviewer kindly suggests, we have incorporated the following sentence; However, patients with immune-mediated inflammatory diseases including psoriasis treated with cytokine inhibitors had reduced susceptibility to SARS-CoV-2 infection compared with patients not receiving cytokine inhibitors, as well as the general population (line 172). In addition, we have revised the conclusion section (line 235).

Reviewer 2 Report

The authors reviewed all published data about the outcome and risk of patients with psoriasis during the COVID19 pandemic. They discussed the advantages and potential  risks of biologics. Recommandations and observed practice for initiation, continuation and/or discontinuation of biologis during the COVID 19 pandemic are described.

This review is well written and useful. Most publications confirmed that biologics did not increase the risk of  COVID19 infection and the development of more severe infection. Some biologicas could bring advantages against the cytokine storm. Risk factors for the development of severe infection were the same as compared to patient without psoriasis. The discontinuation of treatment was associated with the flare of psoriasis. But they also concluded that today there is no clear answer on the benefit or risk of biologics for psoriasis. Decision making needs a careful and personalized  discussion between the physician and the patient.

Author Response

The list of answers for reviewer’s comments

  We are grateful for your critical evaluation of our manuscript and for the detailed evaluation. We are also grateful for the contributions obtained from the reviewers and the opportunity to improve the quality of our manuscript.

Comments and Suggestions for Authors

The authors reviewed all published data about the outcome and risk of patients with psoriasis during the COVID19 pandemic. They discussed the advantages and potential risks of biologics. Recommandations and observed practice for initiation, continuation and/or discontinuation of biologis during the COVID 19 pandemic are described.

This review is well written and useful. Most publications confirmed that biologics did not increase the risk of COVID19 infection and the development of more severe infection. Some biologicas could bring advantages against the cytokine storm. Risk factors for the development of severe infection were the same as compared to patient without psoriasis. The discontinuation of treatment was associated with the flare of psoriasis. But they also concluded that today there is no clear answer on the benefit or risk of biologics for psoriasis. Decision making needs a careful and personalized discussion between the physician and the patient.

Response: We thank the reviewer for favorably reviewing our manuscript. Another reviewer introduced the following article; Simon D, et al. Patients with immune-mediated inflammatory diseases receiving cytokine inhibitors have low prevalence of SARS-CoV-2 seroconversion. Nat Commun. 2020;11(1):3774. In this study, patients with immune-mediated inflammatory diseases including psoriasis treated with cytokine inhibitors had reduced susceptibility to SARS-CoV-2 infection compared with patients not receiving cytokine inhibitors, as well as the general population. In the new manuscript, we conclude that biologics seem to be beneficial to COVID-19 infection (line 237).

Round 2

Reviewer 1 Report

the authors has made corrections according to the previous suggestions.

the paper is now acceptable for publication.

This manuscript is a resubmission of an earlier submission. The following is a list of the peer review reports and author responses from that submission.